# Twenty Novel MicroRNAs in the Aqueous Humor of Pseudoexfoliation Glaucoma Patients

**DOI:** 10.3390/cells12050737

**Published:** 2023-02-24

**Authors:** Marcin Czop, Karolina Gasińska, Ewa Kosior-Jarecka, Dominika Wróbel-Dudzińska, Janusz Kocki, Tomasz Żarnowski

**Affiliations:** 1Department of Clinical Genetics, Medical University of Lublin, 20080 Lublin, Poland; 2Department of Diagnostics and Microsurgery of Glaucoma, Medical University of Lublin, 20079 Lublin, Poland

**Keywords:** miRNA, aqueous humor, pseudoexfoliation glaucoma, PEXG, glaucoma, pseudoexfoliation syndrome, PEX

## Abstract

The microRNAs (miRNAs) are short non-coding RNAs (19–25 nt) that regulate the level of gene expression at the post-transcriptional stage. Altered miRNAs expression can lead to the development of various diseases, e.g., pseudoexfoliation glaucoma (PEXG). In this study, we assessed the levels of miRNA expression in the aqueous humor of PEXG patients using the expression microarray method. Twenty new miRNA molecules have been selected as having the potential to be associated with the development or progression of PEXG. Ten miRNAs were downregulated in PEXG (hsa-miR-95-5p, hsa-miR-515-3p, hsa-mir-802, hsa-miR-1205, hsa-miR-3660, hsa-mir-3683, hsa -mir-3936, hsa-miR-4774-5p, hsa-miR-6509-3p, hsa-miR-7843-3p) and ten miRNAs were upregulated in PEXG (hsa-miR-202 -3p, hsa-miR-3622a-3p, hsa-mir-4329, hsa-miR-4524a-3p, hsa-miR-4655-5p, hsa-mir-6071, hsa-mir-6723-5p, hsa-miR-6847-5p, hsa-miR-8074, and hsa-miR-8083). Functional analysis and enrichment analysis showed that the mechanisms that can be regulated by these miRNAs are: extracellular matrix (ECM) imbalance, cell apoptosis (possibly retinal ganglion cells (RGCs)), autophagy, and elevated calcium cation levels. Nevertheless, the exact molecular basis of PEXG is unknown and further research is required on this topic.

## 1. Introduction

Glaucoma is a group of multifactorial eye diseases that result in progressive irreversible damage to the optic nerve and cause vision loss, and is considered the second leading cause of blindness in the world [1]. The basic process in the pathophysiology of glaucoma is the loss of retinal ganglion cells (RGCs) and their axons, which leads to changes in the morphology of the optic disc and consequently to visual field defects. The main risk factor for glaucoma is increased intraocular pressure (IOP), caused by an imbalance between the production and outflow of aqueous humor (AH) from the anterior chamber [2]. The most important outflow pathway is trabecular meshwork (TM), which consists of trabecular cells surrounded by the extracellular matrix (ECM). Dysfunction in the normal homeostatic process leads to increased outflow resistance and elevated IOP [3]. Lowering IOP is the only treatment that prevents the progression of vision loss [4].

Pseudoexfoliation glaucoma (PEXG) is a type of secondary open-angle glaucoma, developing usually over 50 years of age in the course of pseudoexfoliation syndrome (PEX). The prognosis is worse than for primary open-angle glaucoma (POAG) due to higher IOP values and greater fluctuations. PEX was first described in 1917 by Finnish ophthalmologist, John Linberg [5,6]. It’s a chronic, age-related systemic disorder, with variable prevalence among men and women in different studies [7,8,9,10,11]. Pathognomonic eye symptoms are gray-white deposits in the anterior segment of the eyeball (corneal endothelium, trabecular meshwork, pupillary margin, iris pigment epithelium, dilator muscle of the iris, iris blood vessels, ciliary epithelium, lens epithelium, lens capsule, zonules) and periorbital tissues (conjunctiva, Tenon’s capsule, orbital septa, extraocular muscles) [12,13,14,15]. PEX material can also be found in many internal organs (heart, lungs, liver, kidneys, gallbladder, blood vessels, cerebral meninges) and skin [5]. The deposits are the result of increased production or reduced turnover of ECM [16]. They are composed of elastic fibers (elastin, tropoelastin, amyloid P, fibrillin-1, fibulin-2, vitronectin, fibronectin, lysyl oxidase, clusterin, LTBP-1, LTBP-2, and other proteins) and noncollagenous basement membrane materials (laminin) which form fibrils. They are coated with glycosaminoglycans (heparan sulfate, chondroitin sulfate, dermatan sulfate, and hyaluronic acid) [17,18]. Mechanisms leading to the development of PEX may include oxidative stress, hypoperfusion, and hypoxia [19]. Previous studies revealed many genes that can be related to PEX (*APOE*, *CACNA1A*, *CLU*, *CNTNAP2*, *FBLN5*, *GST*, *LOXL1*, *MMP1*, *MMP3*, *TNF*, *TMEM136*) by participating in the formation of PEX material, regulation of trabecular cells and ECM proteins [18,20,21,22,23]. Gene expression is a complex process in which microRNAs (miRNAs) are involved. Many miRNAs show a tissue-specific expression pattern and can be identified in eye tissues and fluids, but the role of miRNAs in glaucoma and PEXG is not well understood [24,25,26].

MiRNAs are short non-coding RNAs (19–25 nt) that regulate the level of protein expression at the post-transcriptional level. These molecules are responsible for the regulation of many physiological cellular processes, including division, cell differentiation, and regeneration, but also for many pathological processes such as neoplastic transformation. MiRNAs are found both in the cell and in extracellular fluids, i.e., plasma, urine, and aqueous humor [27,28,29,30]. MiRNAs are transcribed in a similar way to protein genes. The first step in the expression of a gene encoding miRNA is the synthesis of primary miRNA (several hundred nucleotides long), which are then modified in the nucleus at both ends of the RNA. The next stage of miRNA maturation takes place in the cytoplasm and, as a consequence, a pre-miRNA of about 70 nucleotides is formed, followed by the production of mature micro RNA. The functioning of micro RNA is closely related to the RNA-induced silencing complex (RISC). Both molecules linked together to interact with the target mRNA through partial complementarity of the sequence of one of the miRNA strands, leading to degradation of the target mRNA through the activity of the RISC complex. By analyzing the level of miRNA expression, it is possible to draw conclusions about the pathomechanism of the studied diseases and to use them as biomarkers of these diseases [27,31,32]. The most important features of biomarkers are: specificity for the disease entity, the possibility of quick, easy, and safe determination, and the possibility of obtaining accurate results [33,34,35,36].

The aim of the study was to assess the levels of miRNA expression in a group of patients with PEXG and to look for mechanisms caused by altered expression of miRNA that may be related to this disease.

## 2. Materials and Methods

### 2.1. Study Groups

Eighteen western descent patients underwent routine cataract surgery at the Department of Diagnostics and Microsurgery of Glaucoma, Medical University of Lublin (Poland) (nine PEXG and nine age-matched control patients). Prior to the surgery, all of the patients underwent the ophthalmic examination including best-corrected visual acuity (BCVA) assessed with Snellen charts, an IOP test measured by Goldmann applanation tonometry (GAT), gonioscopy, a slit-lamp examination, indirect ophthalmoscopy after pupil dilation with a stereoscopic optic nerve head assessment, optical coherence tomography (Carl Zeiss Cirrus HD-OCT 5000), and standard automated perimetry (SAP) (24-2 strategy using the Humphrey perimeter).

Inclusion criteria in the study group:(1)Incipient senile cataract: cloudy area in the lens that causes decreased vision.(2)PEX syndrome: typical deposits of white powdery material observed along the pupillary margin and on the peripheral lens capsule.(3)Glaucomatous optic neuropathy: elevated IOP of greater than 21 mmHg, documented in medical history; open-angle grade III/IV according to Schaffer’s classification; glaucomatous optic nerve head damage (excavation, neuroretinal rim thinning or notching, and localized or diffuse retinal nerve fiber layer [RNFL] defect); and glaucomatous defect in SAP in at least two consecutive tests, with three reliability indices better than 15% (results were considered abnormal if the Glaucoma Hemifield Test result was outside normal limits and at least three contiguous points were present within the same hemifield on the pattern deviation [PD] plot at *p* < 1%, with at least 1 point at *p* < 0.5%).(4)Advanced stages of PEXG: severe visual field loss with MD in visual field test > −18.0 dB.

Inclusion criteria in the control group:(1)Incipient senile cataract: cloudy area in the lens that causes decreased vision.(2)No clinical signs of glaucoma, nor PEX.(3)Normal IOP: from 10 mmHg to 21 mmHg.

In both groups, other causes of RNFL thinning (i.e., myopia, optic disc anomalies, ischemic optic neuropathy, optic neuritis, hereditary optic neuropathy, traumatic optic neuropathy, multiple sclerosis, and degenerative diseases such as Alzheimer’s and Parkinson’s disease) were excluded. Moreover, all of the participants had no previous intraocular surgeries, no other systemic or ocular diseases known to affect the visual field (e.g., pituitary lesions, demyelinating diseases, diabetes mellitus, etc.), and no previous eye or head trauma in their medical history.

The patients’ demographic and clinical data are presented in Table 1.

### 2.2. Aqueous Humor

Approximately 100 µL of AH were obtained from the eye’s anterior chamber at the beginning of the cataract surgery with special care to avoid contamination with blood or tears.

### 2.3. RNA Isolation

The total RNA was isolated from the AH samples using an miRNeasy Serum/Plasma Kit (Qiagen, Valencia, CA, USA) according to the manufacturer’s instructions. The isolated RNA was stored at −80 °C for further analysis. The RNA concentration was determined by using a NanoDrop 2000c spectrophotometer (Thermo Fisher Scientific, Waltham, MA, USA). In addition, RNA analysis was performed using an Agilent Bioanalyzer 2100 (Agilent Technologies, Santa Clara, CA, USA) and a Pico RNA Kit according to the manufacturer’s protocol.

### 2.4. The miRNA Profiling

A microarray system (GeneChip miRNA 4.0 Array chip, Affymetrix, Santa Carla, CA, USA) was used to determine the miRNA expression profiles. The RNA preparation and hybridization were performed according to the manufacturer’s protocol, with one modification to extend the hybridization time to 42 h. The gene chips were scanned with an Affymetrix GeneChip Scanner 3000 (Affymetrix, Santa Carla, CA, USA).

The raw data, in a CEL format, were analyzed as log2-transformed intensities using the Affymetrix Transcriptome Analysis Console (TAC), following the software’s guidelines to determine differentially expressed genes (DEGs) between the PEXG and control patients. The miRNAs that showed at least a one-fold difference between the control and PEXG groups, with a *p*-value of less than 0.01, were considered statistically significant and differentially expressed.

### 2.5. Enrichment Analysis

The functional analysis of selected miRNA was performed using the Diana mirPATH v3.0 (https://dianalab.e-ce.uth.gr/html/mirpathv3/index.php?r=mirpath (accessed on 2 September 2022)) [37]. The visualization of the regulatory network with obtained interactions was carried out by using Cytoscape v3.9.1 software (https://cytoscape.org (accessed on 2 September 2022)) [38].

The analysis of the interactions of the selected miRNAs with genes was carried out in an R environment (version 4.1.2, https://www.r-project.org (accessed on 1 December 2021)) using the multiMiR 1.16.0 package (https://bioconductor.org/packages/release/bioc/html/multiMiR.html, (accessed on 2 September 2022)) [39]. The enrichment analysis was performed using the mirNET database (https://www.mirnet.ca/miRNet/home.xhtml (accessed on 2 September 2022)), where a list of selected miRNAs and a list of genes with which they interact were introduced [40]. Terms of the Kyoto Encyclopedia of Genes and Genomes (KEGG), REACTOME, and Gene Ontology (GO) categories were searched for in PEX-related pathways.

The plot visualizing the enrichment analysis was generated using the ggplot2 3.3.0 package in the R environment.

### 2.6. PPI Network and Subnetwork Construction

Cytoscape was also used to perform protein–protein interactions (PPI). StringApp v2.0 (https://apps.cytoscape.org/apps/stringapp (accessed on 5 January 2023)) [41] was used for this purpose, which uses the Search Tool for the Retrieval of Interacting Genes (STRING). Molecular Complex Detection v2.0.2 (MCODE) (https://apps.cytoscape.org/apps/mcode (accessed on 7 January 2023)) [42] and cytoHubba v0.1 (https://apps.cytoscape.org/apps/cytohubba (accessed on 7 January 2023)) [43] plugins were then used to create clusters and extracting hub genes in a PPI network.

### 2.7. Statistical Analysis

Results at *p* < 0.05 were considered statistically significant. The following software was used: GraphPad Prism 9 (Graph Pad Software, San Diego, CA, USA) and Statistica 13.3 (StatSoft, Krakow, Poland). The analyzes included the Shapiro–Wilk test (to assess the compliance of the examined variables with the normal distribution), the student’s *t-* test, and ROC curves (area under the curve (AUC) were calculated) for variables on a quantitative scale (data presented as an average). Data on a qualitative scale are presented as numbers and percentages and were analyzed using the Chi^2 test.

## 3. Results

Analysis comparing the expression level of miRNAs using expression microarrays in the PEXG group compared to the cataract group showed 20 miRNAs as DEG’s.

Ten miRNAs were downregulated in PEXG (hsa-miR-95-5p, hsa-miR-515-3p, hsa-mir-802, hsa-miR-1205, hsa-miR-3660, hsa-mir-3683, hsa -mir-3936, hsa-miR-4774-5p, hsa-miR-6509-3p, and hsa-miR-7843-3p) (Table 2, Figure 1) and ten miRNAs were upregulated in PEXG (hsa-miR-202-3p, hsa-miR-3622a-3p, hsa-mir-4329, hsa-miR-4524a-3p, hsa-miR-4655-5p, hsa-mir-6071, hsa-mir-6723-5p, hsa-miR-6847-5p, hsa-miR-8074, and hsa-miR-8083) (Table 3, Figure 2).

In the group of DEG’s miRNAs whose expression was lower in PEXG compared to the group (*p* < 0.01; AUC > 0.852; *p* < 0.01), a mean reduction in the level of PEXG expression was found at the level of 29.31% for hsa-miR-95-5p, 36.86% for hsa-miR-515-3p, 29.13% for hsa-mir-802, 28.35% for hsa-miR-1205, 13.15% for hsa-miR-3660, 43.64% for hsa-mir-3683, 42.14% for hsa-mir-3936, 14.35% for hsa-miR-4774-5p, 22.13% for hsa-miR-6509-3p, and 27, 37% for hsa-miR-7843-3p.

In the group of DEG’s miRNAs whose expression was higher in PEXG compared to the group (*p* < 0.01; AUC > 0.840; *p* < 0.01), a mean increase in the level of PEXG expression was found at the level of 28.73% for hsa-miR-202-3p, 29.65% for hsa-miR-3622a-3p, 20.05% for hsa-mir-4329, 21.64 for hsa-miR-4524a-3p, 34.06% for hsa-miR-4655 -5p, 23.50% for hsa-mir-6071, 24.61% for hsa-mir-6723-5p, 24.30% for hsa-miR-6847-5p, 21.57% for hsa-miR-8074, and 23.06% for hsa-miR-8083.

Using the DIANA miRPath v3.0 database, a functional analysis of each of the selected miRNAs was performed (the analysis was performed separately for miRNAs with reduced and increased expression in PEXG). Next, networks were constructed visualizing possible functional relationships between the selected miRNAs (Figure 3 and Figure 4).

The functional network for downregulated miRNA functionally linked together nine miRNAs (leaving one type of miRNA without functional linkage—hsa-miR-3683). The terms with the most associations between the selected miRNAs were: “Ion binding”, “organelle”, “biosynthetic process”, “cellular nitrogen compound metabolic process”, “molecular function” and “molecular function”.

Functional network for upregulated miRNA functionally linked eight miRNAs (leaving two types of miRNAs (has-miR-6723-5p, and hsa-miR-4655-5p) without functional links between other miRNAs and with each other). The terms that obtained the most connections between the selected miRNAs were: “Ion binding”, “organelle”, “biosynthetic process”, “cellular nitrogen compound metabolic process”, “steroid hormone biosynthesis” and “gene expression”.

The next step was to perform an enrichment analysis to evaluate the biological processes regulated by the selected miRNAs and interacted genes using the mirNET database. Five categories were selected: KEGG (Kyoto Encyclopedia of Genes and Genomes), GO: Biological Processing (GO:BP), GO:Cellular Compartment (GO:CC), GO: Molecular Function (GO:MF), and Reactome. Genes indicated as targets of the expressed miRNAs were associated with intracellular and extracellular structures in addition to various processes. The top 10 terms for each category that may be potentially related to glaucoma and/or PEX are presented in Figure 5 and Figure 6, separately for downregulated and upregulated miRNAs in PEXG.

Within the group of downregulated miRNAs, the most frequent terms are related to the regulation of transcription and translation, internal cellular transport (“trans-Golgi network transport vesicle”, “endocytosis”), and signal transduction (Figure 5). In the upregulated miRNA group, the most frequently repeated terms are related to the regulation of transcription and translation, the regulation of the cell cycle, and the activity of cations (mainly calcium and zinc) (Figure 6).

A PPI network of the genes regulated by selected miRNAs were constructed and clustering analysis was performed based on these PPI networks. In total, six clusters (sub-networks) were created for each PPI network of the genes regulated by downregulated miRNAs (Figure 7, Table 4). Clusters I and IV are functionally related to chromatin organization and epigenetic regulation of gene expression, moreover, the key genes in them are the EZH2 gene (in cluster I) and BAZ1B (in cluster IV). Cluster II is functionally related to nuclear chromosome segregation and cellular response to DNA damage (the key gene: TIMELESS). Cluster III is functionally related to the voltage-gated calcium channel complex and MAPK signaling pathway (the key gene: CACNA2D1). Cluster V is functionally related to protein transport (the key gene: RAB7A). Cluster VI is functionally related to ubiquitinization (the key gene: UBE2D1).

In total, six clusters (sub-networks) were created for each PPI network of the genes regulated by upregulated miRNAs (Figure 8, Table 5). Cluster VII is functionally related to cytokine-cytokine receptor interaction and cell–cell adhesion (the key gene: IL10). Cluster VIII is functionally related to the activity of RNA polymerases II and III (the key gene: PLR2D). Cluster IX is functionally related to ubiquitinization (the key gene: PHC1). Cluster X is functionally related to L-type voltage-gated calcium channel complex (the key gene: CACNA1G). Cluster XI is functionally related to the regulation to the jnk cascade (the key gene: AXIN1). Cluster XII is functionally related to Rho GTPases activate ROCKs and phospholipase D signaling pathway (the key gene: RHOA).

The hub genes identified using the cytoHubba app included the EZH2, H2AFX in hub I, CACNA1D in hub IIa, and MYC in hub IIb (Figure 9). Enrichment analysis for these networks indicated that hub I is related to generic transcription pathway and cellular responses to stress, hub IIa is related to voltage-gated channel activity and MAPK signaling pathway, hub IIb is related to cellular senescence and Wnt, Hippo, JAK-STAT signaling pathways (Table 6).

## 4. Discussion

MiRNA molecules are found both inside the cell and in extracellular fluids. The miRNA in the AH may be derived from blood plasma and may be secreted by intraocular cells. These molecules have been shown to be of great importance in processes such as retinal homeostasis and its development [44,45].

Current knowledge about the pathological mechanisms of PEXG is still unknown. Many studies suggest the role of non-coding RNAs, which, through altered expression levels, may be associated with the development of PEXG. The role of miRNAs [26,46,47,48] and snoRNA [49] is mainly suggested.

Current literature suggests a role for certain miRNAs in the pathogenesis of PEXG. These are hsa-miR-671, hsa-miR-374a-5p, hsa-miR-1307-5p, hsa-miR-708-5p [48], and hsa-miR-30d-5p, hsa-miR-320a, hsa- miR-3156-5p, hsa-miR-4458, hsa-miR-6717-5p, hsa-miR-6728-5p, hsa-miR-6834-5p, hsa-miR-6864-5p, hsa-miR-6879- 5p, hsa-miR-877-3p, hsa-miR-548e-3p, and hsa-miR-6777-5p [26]. The main mechanisms associated with the altered expression of these miRNAs is cell apoptosis by suppressing the Wnt3a/β-catenin pathway and the PI3K/AKT pathway [48] and the TGF-beta signaling pathway [26].

In our work, we examined the level of miRNA expression in a group of patients with PEXG and cataracts. After selecting the 20 DEGs of miRNAs (10 upregulated and 10 downregulated), we performed an enrichment and functional analysis.

One of the pathological mechanisms related to PEX contains too high of a concentration of calcium cations, which is directly related to the activity of the channels for these cations. Currently, three types of calcium channels are cited, i.e., CACNA1A [21], CACNB3 [46,49,50], and CACNB4 [50]. The calcium channel is composed of subunits, of which the α subunit forms a transmembrane channel, while the remaining subunits (β and α2/δ) perform regulatory functions. Additionally, in some channels, there may be a γ subunit. The calcium channels are divided into three subfamilies, i.e., CaV1, CaV2, and CaV3. The activity of the CaV1 subfamily channels are responsible for the initiation of contraction and regulates gene expression and transmission in synapses. The CaV2 subfamily channels are associated with synaptic transmission, while the CaV3 subfamily channels are associated with the production of action potentials in myocytes and neurons. The CaV1.4 transporter has been shown to be present in the retinal cell membranes of the eye and its activity is related to visual signaling [51,52]. The results of functional analysis in our study showed that hsa-miR-8074 influences the transport of calcium cations across biological membranes through the regulation of the *ATP2B2* gene. Moreover, we found that the voltage-gated calcium channel activity is being regulated by the genes targeted by selected miRNAs. Additionally, the Wnt signaling pathway is regulated by the genes targeted by selected miRNAs, where one of the signaling pathways is the Wnt/Ca2 + pathway, the activity of which is related to G proteins and phospholipases.

An important regulatory structure of the IOP is HTM (human trabecular meshwork), which is made up of trabecular cells surrounded by the ECM. HTM is responsible for the outflow of AH—in a situation of difficult outflow, the IOP increases [53,54]. The literature provides two pathways (TGFβ1 and TGFβ2), in which the changes lead to an imbalance in the eye’s ECM and, consequently, to an increase in IOP [53,54]. Among the results of the functional analysis in our work, we obtained the term “ECM receptor interactions” for four miRNAs that are regulated by genes: hsa-mir-802 (COL5A1, LAMC1, FN1, CD47), hsa-mir-3660 (LAMA3), hsa-mir -3662a-3p (ITGB8, SV2B, COL24A1, COL4A6, TNR), and hsa-miR-4774-5p (COL11A1, COL24A1). In addition, proteins are a very important element of the ECM. Disruption of protein metabolism (synthesis, post-translational modification, or transport) may lead to the development of PEX [55].

Functional analysis of selected miRNAs showed that a relatively large number of these miRNAs influence genes related to protein metabolism. The process of protein modification in the cell is regulated by genes whose expression is influenced by six out of twenty selected miRNAs (hsa-miR-95-5p, hsa-miR-1205, hsa-miR-3660, hsa-miR-515-3p, and hsa- miR-6071,22-3p). In addition, subsequent miRNAs affect the expression level of genes related to protein metabolism, i.e., “protein complex” (hsa-miR-4329, hsa-miR-6071, hsa-miR-1205, and hsa-miR-3660), “cytoskeletar protein binding” (hsa-miR-6847-5p, 1205), “post-translational protein modification” (hsa-miR-7843-3p), “endoplasmic reticulum-Golgi intermediate compartment” (has-miR-802), and “cytoplasmic sequestering of protein “(Hsa-miR-4655-5p). Moreover, hsa-miR6723-5p regulates the expression level of the UBE2H gene related to ubiquitin-mediated proteolis. Among the enriched functional terms of genes targeted by selected miRNAs in AH in PEXG patients, it is worth mentioning the “trans-Golgi network transport vesicle” as a process that influences protein transport. In addition, many times the results of this analysis were terms related to the process of protein biosynthesis (“positive regulation of translation”, “regulation of translation”) and transcription (“negative regulation of transcription from RNA polymerase II promoter”, “negative regulation of transcription, DNA- dependent “). In a previous work, we suggested that some snoRNA molecules are related to PEXG [49]. One of the terms obtained in enrichment analysis is “spliceosomal complex” which may suggest that the selected miRNAs may also be responsible for this process.

The mechanisms discussed above largely explained where the elevated IOP comes from. Elevated IOP results in RGC apoptosis and, furthermore, optic nerve degeneration which is often observed in late PEXG [56,57,58]. Another mechanism that can lead to IOP increase and cell death in glaucoma is oxidative stress [59,60,61,62]. The hsa-miR-802 molecule by regulating the expression of MCL1, PPP2CB, and SOD2 genes influences the processes taking place in the mitochondria during apoptosis. Among the enriched functional terms of genes targeted by selected miRNAs in AH in PEXG patients, it is worth mentioning “G0 and early G1” and “G2/M transition of the mitotic cell cycle”. In addition, “AKT phosphorylates targets in the cytosol” and “PI3K/AKT activation” are terms that suggest that selected miRNAs may regulate the survival of ocular neurons and thus may lead to optic nerve degeneration. The role of the PI3K/Akt pathway was previously mentioned [63,64,65,66,67].

Hsa-miR-802 shows an increased expression in primary human umbilical vein endothelial cells (HUVECs) in response to induced hypoxia [67]. As above, hsa-miR-202-3p is one of the miRNAs whose expression levels have been altered in lung adenocarcinoma A549 cells by hypoxia [68].

Additionally, a correlation between the risk of developing normal-tension glaucoma (NTG) and the level of retinol in the blood serum was demonstrated [69]. Moreover, another group of scientists reports that retinoic acid (a retinol derivative) lowers the expression level of the gene which codes the small heat shock protein B8 (*HSPB8*) [70], which is a chaperone protein functionally related to chaperone-assisted selective autophagy (CASA) [71]. One of the miRNA molecules selected as DEGs in our work, hsa-miR-6509-3p regulates the expression level of the *HSPB8* gene. The above data may suggest a potential influence of the autophagy process in the development of PEXG.

The results of the PPI analysis suggest and confirm that a very important mechanism of PEXG may be disturbances in the transport of calcium cations in the eye. One of the hub genes that have been selected is the CACNA1D gene.

Our research had some limitations. First, the study groups were small, consisting of nine patients each. A study involving a larger group of patients is needed to confirm the results. Furthermore, PEXG patients had advanced neuropathy, which suggests that miRNA molecules may be the result of the pathological process and not a causative factor. However, we precluded that the same stage of the disease within the group would enable more unique results. Since the advanced stages of PEXG, comparison of miRNAs in the early stages of PEXG is needed. In addition, all enrolled patients were diagnosed with senile cataracts. Therefore, the impact of cataracts on the expression of miRNA in the PEXG group should be taken into consideration, but in the studied group all patients also had similar stages of cataracts. However, it is ethically impossible to collect aqueous humor in healthy eyes due to the invasive nature of the procedure.

Until now, the scientific literature has not provided such information, which means that this is the first time we are doing it. These mechanisms are possible, but their role in the pathogenesis of PEXG/PEX must be carefully studied.

## 5. Conclusions

Our results suggest 20 new miRNAs whose altered expression in the aqueous humor may have an influence on the pathomechanisms of PEXG. Functional and enrichment analysis confirms the mechanisms, such as high levels of calcium cations, ECM imbalance, apoptosis of RGC cells, and the optic nerve damage or autophagy as possible causes of PEXG. Moreover, the PPI analysis suggests that CACNA1D, EZH2, and MYC may be hub genes for PEXG. However, confirmation of these conclusions requires further study.

## Figures and Tables

**Figure 1 cells-12-00737-f001:**
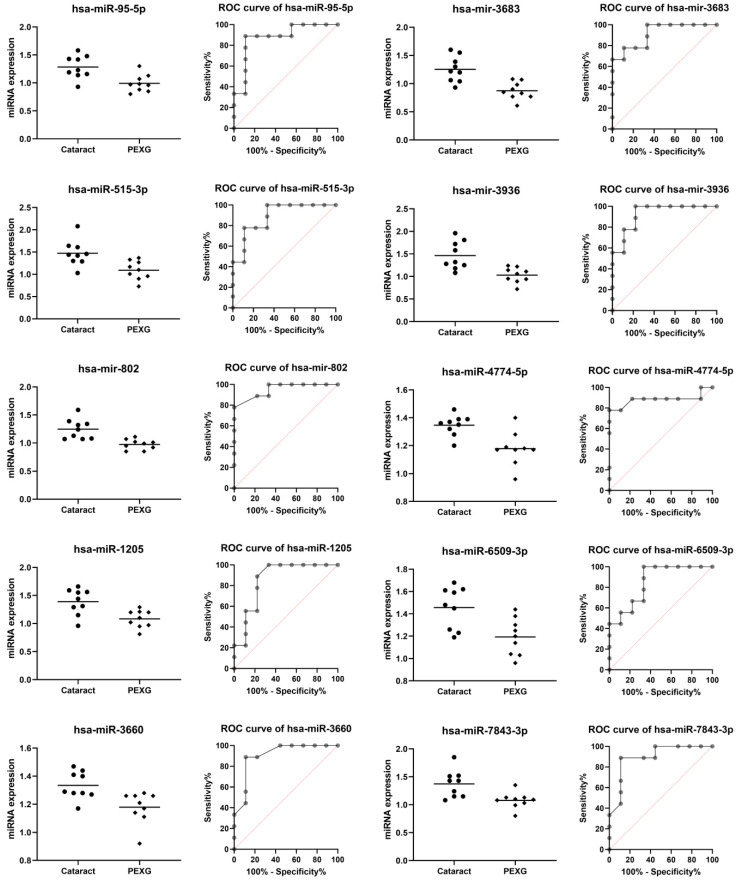
Lower miRNA expression in the PEXG group compared to the cataract group. Data are shown as mean [log2].

**Figure 2 cells-12-00737-f002:**
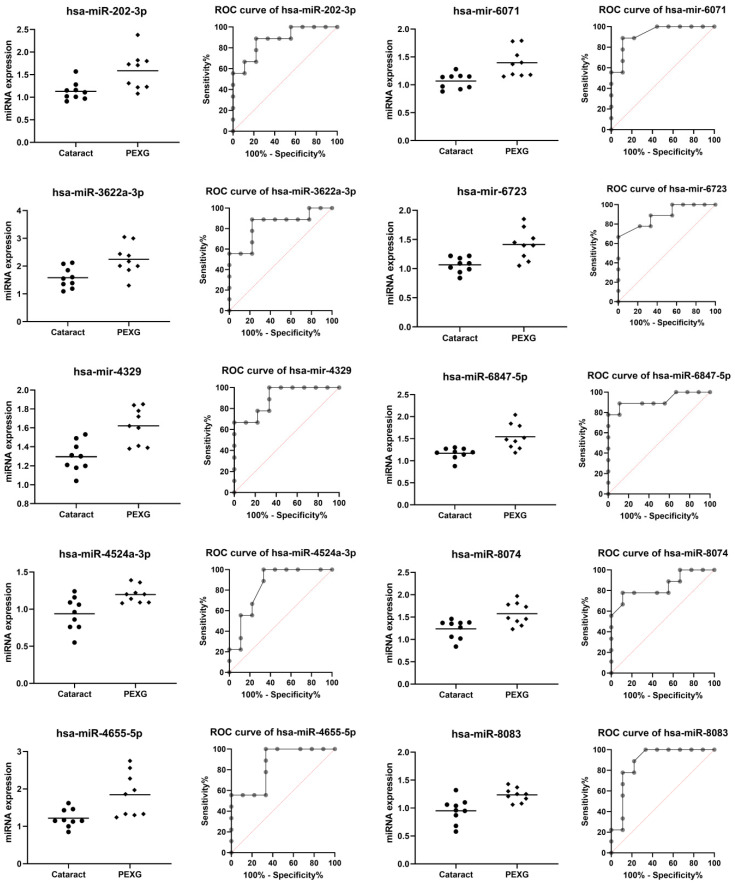
Higher miRNA expression in the PEXG group compared to the cataract group. Data are shown as mean [log2].

**Figure 3 cells-12-00737-f003:**
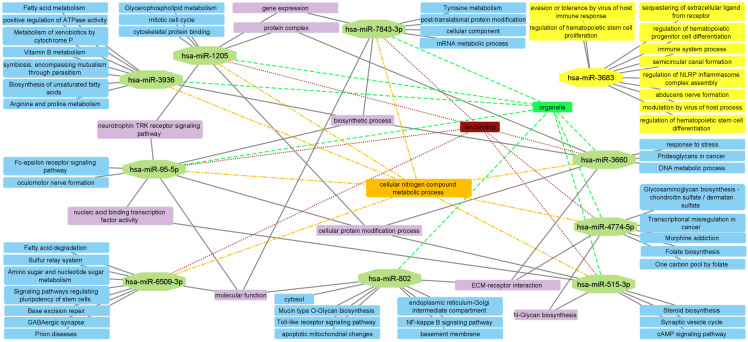
Functional network performed using selected downregulated miRNAs in PEXG. The network is made up of the 10 most statistically significant for each miRNA. Analysis made using DIANA-miRPath v3.0.

**Figure 4 cells-12-00737-f004:**
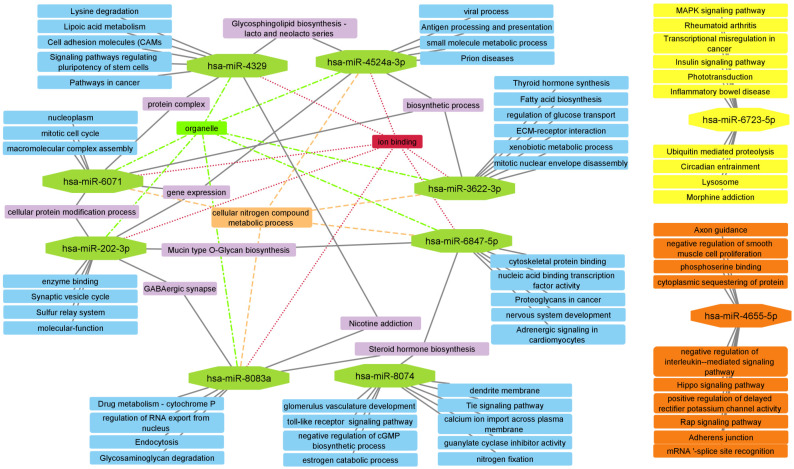
Functional network performed using selected upregulated miRNAs in PEXG. The network is made up of the 10 most statistically significant for each miRNA. Analysis made using DIANA-miRPath v3.0.

**Figure 5 cells-12-00737-f005:**
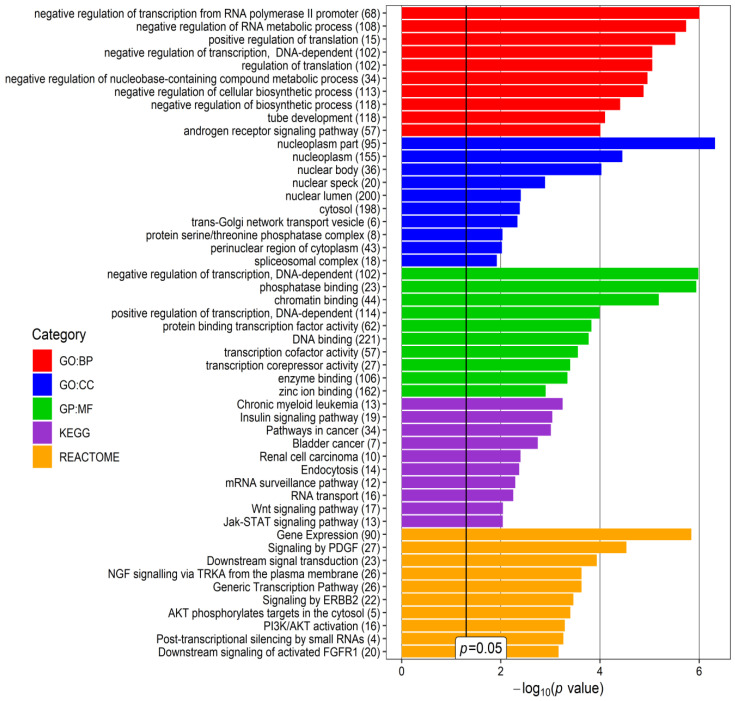
Top 10 terms of the Gene Ontology (GO): Biological Processing (GO:BP), GO: Cellular Compartment (GO:CC), Molecular Function (GO:MF), REACTOME and KEGG (Kyoto Encyclopedia of Genes and Genomes) categories, revealed for glaucoma- and/or pseudoexfoliation syndrome (PEX)-associated genes targeted by downregulated miRNAs found in the current study in the aqueous humor (AH) of patients with PEXG and age-matched controls. *p*-value—EASE score for enrichment adjusted by the Benjamini correction for multiple-hypothesis testing. The number in brackets following the names of terms indicates the number of associated genes. The plot was generated using the ggplot2 3.3.0 package in the R environment.

**Figure 6 cells-12-00737-f006:**
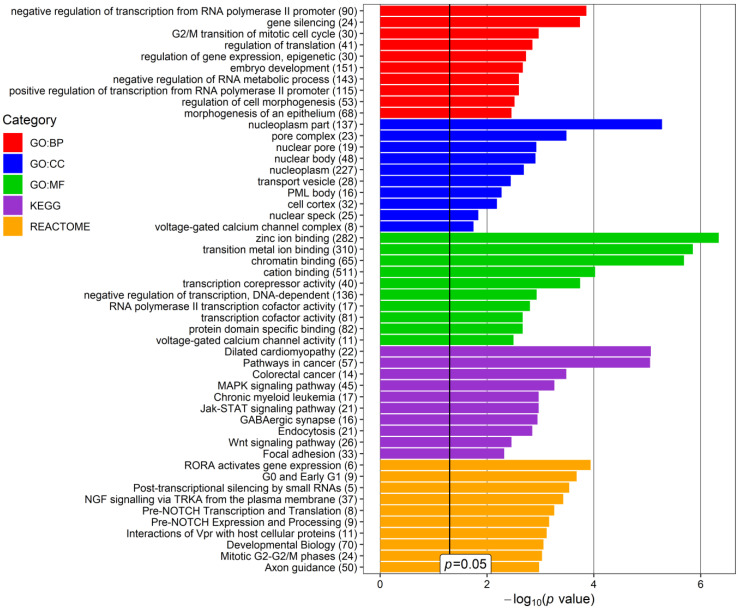
Top 10 terms of the Gene Ontology (GO): Biological Processing (GO:BP), GO: Cellular Compartment (GO:CC), Molecular Function (GO:MF), REACTOME, and KEGG (Kyoto Encyclopedia of Genes and Genomes) categories, revealed for glaucoma- and/or pseudoexfoliation syndrome (PEX)-associated genes targeted by upregulated miRNAs found in the current study in the aqueous humor (AH) of patients with PEXG and age-matched controls. *p*-value—EASE score for enrichment adjusted by the Benjamini correction for multiple-hypothesis testing. The number in brackets following the names of terms indicates the number of associated genes. The plot was generated using the ggplot2 3.3.0 package in an R environment.

**Figure 7 cells-12-00737-f007:**
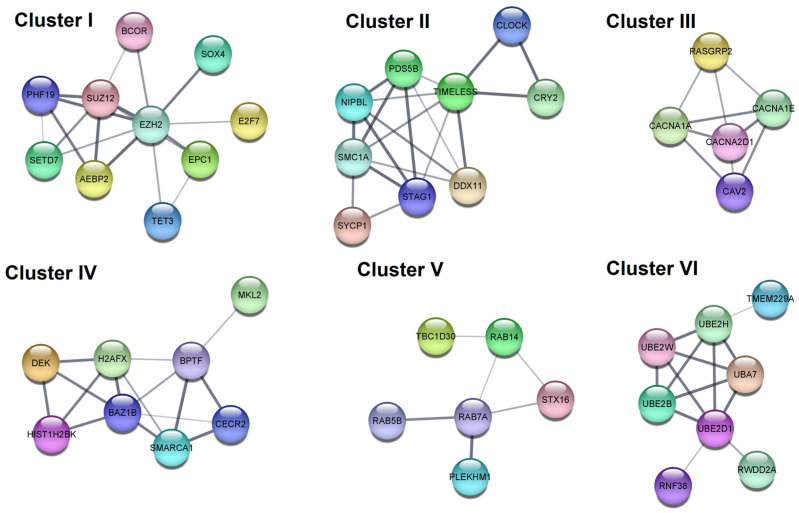
Results of protein–protein interaction analysis. The 6 top-scored clusters separated from the PPI network of the genes regulated by downregulated miRNAs found in the current study. The PPI hub gene module (g).

**Figure 8 cells-12-00737-f008:**
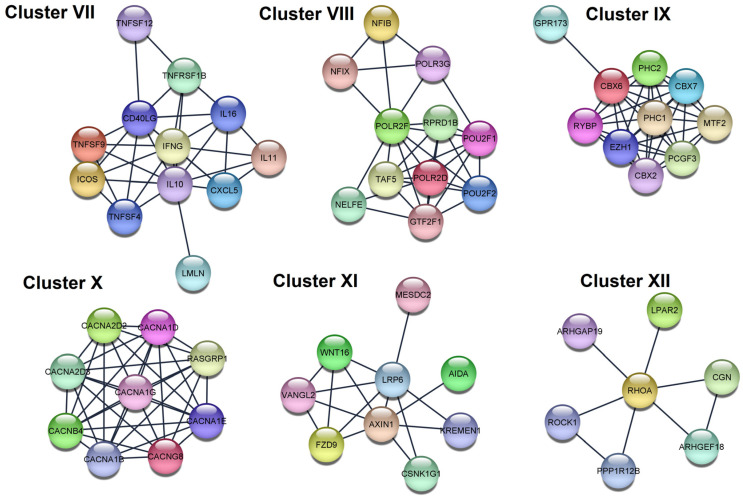
Results of protein–protein interaction analysis. The six top-scored clusters separated from PPI network of the genes regulated by upregulated miRNAs found in the current study.

**Figure 9 cells-12-00737-f009:**
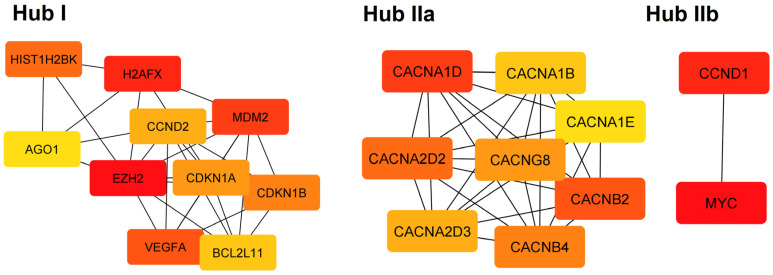
The PPI hub gene modules from PPI networks of the genes regulated by downregulated (Hub I) and upregulated miRNAs (Hubs IIa, IIb).

**Table 1 cells-12-00737-t001:** Demographic and clinical characteristics of the study groups.

		PEXG N = 9	Cataract N = 9	*p*-Value
Age (mean ± SD)		77.56 ± 6.54	76.00 ± 6.95	0.6314 ^^^
Gender (N, %)	Male	8 (88, 89%)	8 (88, 89%)	0.7647 ^#^
	Female	1 (11, 11%)	1 (11, 11%)	
BCVA		0.19 ± 0.17	0.42 ± 0.12	0.0047 ^^^
MAX IOP		30.44 ± 13.1	15.67 ± 2.12	0.0042 ^^^
C/D		0.91 ± 0.07	0.29 ± 0.15	<0.0001 ^^^
RNFL		61.22 ± 5.12	93.8 ± 5.61	0.0005 ^^^
MD		−22.55 ± 4.77	−0.03 ± 0.98	<0.0001 ^^^

^#^—chi^2^ test; ^^^—Student’s *t*-test; BCVA—best-corrected visual acuity; C/D—cup/disc; IOP—intraocular pressure; MD—mean deviation; PEXG—pseudoexfoliation glaucoma; RNFL—retinal nerve fiber layer; and SD—standard deviation.

**Table 2 cells-12-00737-t002:** 10 miRNA downregulated in the PEXG group compared to the cataract group [log2].

miRNA	Group	n	Mean	SD	Fold Change	*p* Value	AUC	AUC *p* Value
hsa-miR-95-5p	Cataract	9	1.284	0.206	−1.25	0.0037	0.877	0.0071
	PEXG	9	0.993	0.155				
hsa-miR-515-3p	Cataract	9	1.474	0.291	−1.25	0.0060	0.889	0.0054
	PEXG	9	1.093	0.213				
hsa-mir-802	Cataract	9	1.248	0.179	−1.20	0.0009	0.951	0.0013
	PEXG	9	0.974	0.091				
hsa-miR-1205	Cataract	9	1.390	0.232	−1.26	0.0046	0.858	0.0104
	PEXG	9	1.083	0.156				
hsa-miR-3660	Cataract	9	1.334	0.099	−1.08	0.0071	0.907	0.0036
	PEXG	9	1.179	0.114				
hsa-mir-3683	Cataract	9	1.254	0.229	−1.30	0.0008	0.914	0.0031
	PEXG	9	0.873	0.153				
hsa-mir-3936	Cataract	9	1.464	0.311	−1.29	0.0020	0.926	0.0023
	PEXG	9	1.030	0.169				
hsa-miR-4774-5p	Cataract	9	1.347	0.074	−1.13	0.0026	0.883	0.0062
	PEXG	9	1.178	0.121				
hsa-miR-6509-3p	Cataract	9	1.457	0.187	−1.21	0.0060	0.852	0.0118
	PEXG	9	1.193	0.165				
hsa-miR-7843-3p	Cataract	9	1.373	0.244	−1.22	0.0066	0.895	0.0047
	PEXG	9	1.078	0.145				

AUC—area under curve.

**Table 3 cells-12-00737-t003:** 10 miRNA upregulated in the PEXG group compared to the cataract group [log2].

miRNA	Group	N	Mean	SD	Fold Change	*p* Value	AUC	AUC *p* Value
hsa-miR-202-3p	Cataract	9	1.131	0.199	1.43	0.0088	0.877	0.0071
	PEXG	9	1.587	0.412				
hsa-miR-3622a-3p	Cataract	9	1.580	0.370	1.55	0.0084	0.840	0.0152
	PEXG	9	2.246	0.552				
hsa-mir-4329	Cataract	9	1.296	0.158	1.25	0.0012	0.901	0.0041
	PEXG	9	1.621	0.191				
hsa-miR-4524a-3p	Cataract	9	0.938	0.223	1.18	0.0069	0.840	0.0152
	PEXG	9	1.197	0.114				
hsa-miR-4655-5p	Cataract	9	1.218	0.242	2.17	0.0088	0.852	0.0118
	PEXG	9	1.847	0.584				
hsa-mir-6071	Cataract	9	1.068	0.137	1.23	0.0037	0.926	0.0023
	PEXG	9	1.396	0.255				
hsa-mir-6723-5p	Cataract	9	1.066	0.131	1.25	0.0026	0.889	0.0054
	PEXG	9	1.414	0.263				
hsa-miR-6847-5p	Cataract	9	1.168	0.128	1.24	0.0025	0.920	0.0027
	PEXG	9	1.543	0.288				
hsa-miR-8074	Cataract	9	1.236	0.211	1.16	0.0070	0.846	0.0134
	PEXG	9	1.576	0.254				
hsa-miR-8083	Cataract	9	0.951	0.222	1.20	0.0040	0.883	0.0062
	PEXG	9	1.236	0.123				

AUC—area under curve.

**Table 4 cells-12-00737-t004:** Enrichment analysis of the six top-scored clusters separated from the PPI network of the genes regulated by downregulated miRNAs found in the current study.

Cluster	Category	Term	*p* Value	Genes
I	GO BP	Chromatin organization	5.00 × 10^−4^	EPC1, SETD7, SUZ12, EZH2, BCOR, AEBP2, TET3, PHF19
	GO BP	Negative regulation of gene expression, epigenetic	0.0012	EPC1, SUZ12, EZH2, AEBP2, PHF19
II	GO BP	Nuclear chromosome segregation	2.70 × 10^−4^	NIPBL, PDS5B, SMC1A, SYCP1, STAG1, DDX11
	GO BP	Cellular response to DNA damage stimulus	0.0023	NIPBL, CLOCK, PDS5B, SMC1A, SYCP1, DDX11, TIMELESS
III	KEGG	MAPK signaling pathway	5.60 × 10^−4^	RASGRP2, CACNA2D1, CACNA1A, CACNA1E
	GO:CC	Voltage-gated calcium channel complex	0.0053	CACNA2D1, CACNA1A, CACNA1E
IV	GO:BP	Chromatin organization	0.0012	BPTF, CECR2, BAZ1B, HIST1H2BK, SMARCA1, DEK, H2AFX
	Reactome	B-WICH complex positively regulates rRNA expression	0.0042	BAZ1B, HIST1H2BK, DEK, H2AFX
V	Reactome	RAB GEFs exchange GTP for GDP on RABs	0.0135	RAB7A, RAB5B, RAB14
	GO:BP	Protein transport	0.0260	RAB7A, RAB5B, STX16, RAB14, PLEKHM1, TBC1D30
VI	GO:MF	Ubiquitin-protein transferase activity	5.60 × 10^−4^	RNF38, UBE2B, UBA7, UBE2H, UBE2D1, UBE2W
	Reactome	Antigen processing: Ubiquitination & Proteasome degradation	0.0013	UBE2B, UBA7, UBE2H, UBE2D1, UBE2W

**Table 5 cells-12-00737-t005:** Enrichment analysis of the six top-scored clusters separated from the PPI network of the genes regulated by upregulated miRNAs found in the current study.

Cluster	Category	Term	*p* Value	Genes
VII	KEGG	Cytokine-cytokine receptor interaction	4.03 × 10^−14^	IFNG, TNFSF9, IL11, TNFSF4, TNFSF12, CXCL5, IL16, CD40LG, TNFRSF1B, IL10
	GO:BP	Positive regulation of cell–cell adhesion	8.60 × 10^−5^	IFNG, TNFSF9, TNFSF4, ICOS, CD40LG, IL10
VIII	GO:CC	RNA polymerase II, holoenzyme	2.90 × 10^−6^	POLR2D, TAF5, RPRD1B, GTF2F1, POLR2F
	Reactome	RNA Polymerase III Abortive Additionally, Retractive Initiation	4.94 × 10^−6^	POU2F1, NFIB, NFIX, POLR3G, POLR2F
IX	Reactome	Regulation of PTEN gene transcription	8.30 × 10^−4^	PHC2, CBX2, CBX6, PHC1
	GO:BP	Histone ubiquitination	0.021	PCGF3, RYBP, PHC1
X	GO:CC	Voltage-gated calcium channel complex	1.29 × 10^−17^	CACNG8, CACNA1D, CACNA1G, CACNA1E, CACNA1B, CACNA2D2, CACNA2D3, CACNB4
	GO:CC	L-type voltage-gated calcium channel complex	0.0021	CACNG8, CACNA1D
XI	KEGG	Basal cell carcinoma	7.70 × 10^−4^	WNT16, AXIN1, FZD9
	GO:BP	Regulation of jnk cascade	0.0119	WNT16, AXIN1, AIDA, VANGL2
XII	Reactome	RHO GTPases Activate ROCKs	5.88 × 10^−5^	ROCK1, RHOA, PPP1R12B
	KEGG	Phospholipase D signaling pathway	0.0283	RHOA, LPAR2

**Table 6 cells-12-00737-t006:** Enrichment analysis of the hub gene module from the PPI network of the genes regulated by downregulated and upregulated miRNAs.

Hub	Category	Term	*p* Value	Genes
I	Reactome	Generic Transcription Pathway	9.79 × 10^−7^	CDKN1B, MDM2, CCND2, EZH2, HIST1H2BK, AGO1, BCL2L11, CDKN1A, H2AFX, VEGFA
I	Reactome	Cellular responses to stress	9.79 × 10^−7^	CDKN1B, MDM2, EZH2, HIST1H2BK, AGO1, CDKN1A, H2AFX, VEGFA
I	Reactome	Cellular Senescence	9.79 × 10^−7^	CDKN1B, MDM2, EZH2, HIST1H2BK, AGO1, CDKN1A, H2AFX
I	KEGG	MicroRNAs in cancer	1.55 × 10^−6^	CDKN1B, MDM2, CCND2, EZH2, BCL2L11, CDKN1A, VEGFA
IIa	GO:CC	Voltage-gated calcium channel complex	9.03 × 10^−14^	CACNG8, CACNA1D, CACNB2, CACNA1E, CACNA1B, CACNA2D2, CACNA2D3, CACNB4
IIa	GO:MF	Voltage-gated calcium channel activity	3.10 × 10^−13^	CACNG8, CACNA1D, CACNB2, CACNA1E, CACNA1B, CACNA2D2, CACNA2D3, CACNB4
IIa	KEGG	MAPK signaling pathway	1.92 × 10^−10^	CACNG8, CACNA1D, CACNB2, CACNA1E, CACNA1B, CACNA2D2, CACNA2D3, CACNB4
IIa	Reactome	Presynaptic depolarization and calcium channel opening	4.50 × 10^−9^	CACNB2, CACNA1E, CACNA1B, CACNA2D2, CACNA2D3, CACNB4
IIb	KEGG	Cellular senescence	0.0091	CCND1, MYC
IIb	KEGG	Wnt signaling pathway	0.0091	
IIb	KEGG	Hippo signaling pathway	0.0091	
IIb	KEGG	JAK-STAT signaling pathway	0.0091	

## Data Availability

All data generated or analyzed during this study are included in this published article.

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
