# Peer review of "Twenty Novel MicroRNAs in the Aqueous Humor of Pseudoexfoliation Glaucoma Patients"

_cells, 2023, doi:10.3390/cells12050737_

Round 1

Reviewer 1 Report (Previous Reviewer 1)

The manuscript has been improved as requested. In my opinion, now it is suitable for publication.

Reviewer 2 Report (Previous Reviewer 3)

The manuscript has been significantly improved and now I suggest the publication in Cells.

This manuscript is a resubmission of an earlier submission. The following is a list of the peer review reports and author responses from that submission.

Round 1

Reviewer 1 Report

please spell ECM and other abbreviations both in the abstract and the text the first time they are mentioned

Please change caucasian with western descent 

please improve the statistical analysis section 

Reviewer 2 Report

In this manuscript, the authors demonstrated the 20 novel miRNAs for pseudoexfoliation glaucoma (PEXG). And they found that these miRNAs may regulate some pathways associated with development or progression of PEXG. The manuscript was well writing. However, miRNA studies in PEXG are not novel, and there were already a few miRNA studies for aqueous humor of PEXG patients (Sci Rep. 2022 Apr 13;12(1):6217; Graefes Arch Clin Exp Ophthalmol. 2021 Aug;259(8):2337-2349). Thus, there were already a few miRNA studies for anterior lens capsules/blood in PEXG/PEX (Genes (Basel). 2022 Mar 25;13(4):582; Genes (Basel). 2022 Mar 10;13(3):489ï¼›Adv Clin Exp Med. 2020 Dec;29(12):1399-1405ï¼› Clin Ophthalmol. 2020 Oct 5;14:3025-3038.) Furthermore, in depth functional analysis should be further conducted in this study.

Other major comments were as follows:

1. Other microarray or sequencing study usually showed more than 100 dysregulated miRNAs, but this study only showed 20 novel miRNAs. What is the total numbers of probe in the microarray used in this study? The screen procedure of dysregulated miRNAs should be shown in details. The miRNA profile should be also shown in a heatmap or volcano plot. Details and illustration of ROC curve should be added in the figures or text.

1. PCR should be conducted to further validate the accuracy and consistency of results of miRNA microarray.

2. Demographic of Patients with PEXG and controls should be shown as a table in this manuscript.

3.Authors should add more details about the diagnosis and enrollment criteria of the enrolled participants.

4.MiRNA-mRNA network should be constructed to show the interaction of these miRNAs and targeted mRNAs.

5.Some meaningful pathways associated with PEGX should be listed in the results or marked in the figures. In addition, GO analysis and pathways analysis may only show the GO term or pathway of “apoptosis”. If you want to show the mechanism “RGC cell apoptosis”, authors should conduct an experiment in vivo and/or in vitro to further validate the RGC cell apoptosis and eliminate the misunderstanding. Otherwise, do NOT mentioned “RGC cell apoptosis” in abstract.

6.Potential functions of one or two potential miRNA with high accuracy of diagnosis should be further investigated.

7. In the discussion section, authors may compare their results with other studies for the finding of the miRNAs.

8. Limitations of this study should be illustrated in a paragraph in the discussion section.

Reviewer 3 Report

The manuscript titled ‘Twenty Novel MicroRNAs in the Aqueous Humor of Pseudoexfoliation Glaucoma Patients’ made a thoroughly demonstration of the levels of miRNA expression in the aqueous humor of PEXG patients using the expression microarray method. This work is very important because many miRNAs show a tissue-specific expression pattern and can be identified in eye tissues and fluids, but the role of miRNAs in glaucoma and PEXG is not well understood. The topic is very current due to the significant role of biomarkers.

This is a high-quality, complex work, I just have a few questions and comments.

Major comments:

1. At the end of the introduction you mentioned “biomarkers of these disease”, but you did not write some samples. I would like to read about the characteristics of ideal biomarkers as well.

2. In the ‘Study Groups’ section you wrote: “The demographic and clinical characteristics of the studied patients were given in the previous work [33].” What were these characteristic? The previous work [33] is a study of an other research group (Liu et al., 2018).

3. Based on your result what do you think which miRNAs could be ideal biomarkers of PEXG?

Minor comments:

1. In my opinion the first sentence of the manuscript is too short, I miss a correct definition of glaucoma (i.e. you should write that this is a multifactorial disease as well).

2. Row 135: was used:. Graph Pad Prism 9 (plus point)

3. Row 162: hsa-miR-8074 , (plus space)

4. Row 231: cells. these molecules (These)

5. Row 242 and 246: ie (i.e.)